Resolving taphonomic and preparation biases in silicified faunas through paired acid residues and X-ray microscopy

Jacobs Gabriel S. 1 2
Jacquet Sarah Monique 2
Selly Tara 2 3
http://orcid.org/0000-0003-4726-0355 Schiffbauer James D. 2 3
Huntley John Warren 2 huntleyj@missouri.edu
1 Department of Geology, Cornell College , Mount Vernon, IA , USA
2 Department of Geological Sciences, University of Missouri , Columbia, MO , United States
3 X-ray Microanalysis Laboratory, University of Missouri , Columbia, MO , United States
Badenhorst Shaw
Electronic publication date: 2024 Feb 1
Publication date: 2024
Volume: 12
Electronic Location ID: e16767
Received 2023 Aug 3; Accepted 2023 Dec 15
Copyright: © 2024 Jacobs et al.
Copyright year: 2024
Copyright holder: Jacobs et al.
License: This is an open access article distributed under the terms of the Creative Commons Attribution License, which permits unrestricted use, distribution, reproduction and adaptation in any medium and for any purpose provided that it is properly attributed. For attribution, the original author(s), title, publication source (PeerJ) and either DOI or URL of the article must be cited.
License URL: https://creativecommons.org/licenses/by/4.0/

Keywords: Computed tomography, Silicification, Ordovician, Carbonates, Taphonomy, Paleoecology, Body size

Funding: Department of Geological Sciences, University of Missouri NSF EAR CAREER 1650745 X-ray Microanalysis Lab were supported by NSF Instrumentation and Facilities (EAR/IF) 1636643 NSF Sedimentary Geology and Paleobiology Award (EAR-SGP) 1917031 Gabriel S. Jacobs was supported by the Department of Geological Sciences, University of Missouri. John Warren Huntley and Gabriel S. Jacobs were supported by NSF EAR CAREER 1650745. James D. Schiffbauer, Tara Selly, and the X-ray Microanalysis Lab were supported by NSF Instrumentation and Facilities (EAR/IF) 1636643. Sarah M. Jacquet and Tara Selly were supported by NSF Sedimentary Geology and Paleobiology Award (EAR-SGP) 1917031. The funders had no role in study design, data collection and analysis, decision to publish, or preparation of the manuscript.

==============================
Paired petrography and acid maceration has shown that preferential silicification of shelly faunas can bias recovery based on taxon and body size. Here, silicified fossils from the Upper Ordovician Edinburg Formation, Strasburg Junction, Virginia, USA, were analyzed using X-ray tomographic microscopy (μCT) in conjunction with recovered residues from acid maceration of the same materials to further examine sources of potential bias. Results reveal that very small (<~1 mm) fossils are poorly resolved in μCT when scanning at lower resolutions (~30 µm), underestimating abundance of taxa including ostracods and bryozoans. Acid maceration, meanwhile, fails to recover poorly silicified fossils prone to disarticulation and/or fragmentation during digestion. Tests for patterns of breakage, however, indicate no significant size or taxonomic bias during extraction. Comparisons of individual fossils from 3-D fossil renders and maceration residues reveal patterns of fragmentation that are taxon-specific and allow the differentiation of biostratinomic and preparational breakage. Multivariate ordinations and cluster analyses of μCT and residue data in general produce concordant results but indicate that the variation in taxonomic composition of our samples is compromised by the resolvability of small size classes in μCT imaging, limiting the utility of this method for addressing paleoecological questions in these specific samples. We suggest that comparability of results will depend strongly on the sample size, taphonomic history, textural, and compositional characteristics of the samples in question, as well as μCT scan parameters. Additionally, applying these methods to different deposits will test the general applicability of the conclusions drawn on the relative strengths and weaknesses of the methods.

Introduction

Silicification is a common preservational mode in the fossil record, especially in the early Paleozoic (Butts, 2007; Butts & Briggs, 2010), involving the fabric-specific replacement of calcareous biota (calcite and aragonite) with silica, typically as chalcedony (fine-grained fibrous quartz), opaline silica, or sometimes as sparry (macrocrystalline) quartz cement. While the fine-scale mechanisms of silicification in limestone remain unclear, the overall process is simple: pore fluids containing dissolved silica (whether volcanic, hydrothermal, or biogenic in origin) undergo transport and react with the host rock, causing contemporaneous dissolution of calcite and/or aragonite and the precipitation of silica through mechanisms involving pH change, increased temperature due to burial depth, and pressure dissolution (Kastner, Keene & Gieskes, 1977; Maliva & Siever, 1988).

Due to their enhanced resistance to other fabric-destructive diagenetic process (Schubert, Kidder & Erwin, 1997), silicified fossils are typically preserved in great fidelity and abundance. Moreover, the replacement process frequently replicates fine morphological details if not internal microstructure, and the insolubility of silica to dissolution by acid allows such material to be prepared chemically rather than mechanically (Baars, 2008; Butts, 2014). Acid maceration is a broadly employed technique used to extract silicified fossils from limestone, relying on the high solubility of calcite in acidic solution and the comparatively low resistance of micritic matrix to alteration or dissolution (St. Clair, 1935; Grant, 1989). However, such extraction methods can introduce potential biases that might skew perceptions of paleoecological studies investigating silicified specimens (Pruss, Payne & Westacott, 2015). Two primary sources of bias that can readily be identified include the differential likelihood that a given specimen will (1) silicify during diagenesis, and (2) once silicified, survive preparation and be recovered.

If biases in differential preservation and recovery methods go unaccounted for, certain types of analyses can be more severely affected. Systematic descriptions of organisms may be only mildly affected by the underrepresentation of some ontogenetic stages relative to others, but studies specific to the paleoecology or biofacies of an assemblage can suffer in quality due to their reliance on an accurate assessment of which biota are present in the deposit. For example, the near absence of a guild of predators or the preferential breakage of certain types of shell ornamentation might drastically alter the interpretation of trophic relationships and biotic interactions. Similarly, the loss of fossils in a certain size range during preparation can affect interpretations of biostratinomic processes. Because these biases may skew paleoecological interpretations of fossil assemblages, quantifying or at least constraining them can improve confidence in the results of research relying on this common preparatory method.

Previous research has attempted to account for preservational and preparation bias in acid maceration of silicified fossils. Pruss, Payne & Westacott (2015) compared the relative abundances of fossil taxa in 11 hand samples of the Triassic Virgin Limestone Member of the Moenkopi Formation, from which the authors made a petrographic thin section and extracted fossil specimens via buffered acetic acid maceration. In general, echinoderms and gastropods were disproportionally over-represented in residues while bivalves were more likely to be identified in thin-section point counts (Pruss, Payne & Westacott, 2015).

Two questions not addressed by the aforementioned study are those of body size bias and breakage patterns. Since analysis of petrographic thin sections can identify fossils by their taxonomic group but offers minimal useful information with respect to their dimensions, detecting differences in body size distribution between thin section grains and fossils recovered in residues is not feasible. Similarly, it is impossible to tell by extraction methods whether fossils in residues suffered damage during digestion and washing or whether the breakage occurred prior to burial. Because thin sections and residues do not record the same fossils but rather disjoint subsets of the total fossils within the hand sample, no true pre and post comparison can be made. Indeed, since both methods applied in the Pruss, Payne & Westacott (2015) study are destructive, a single fossil cannot be recorded by both. This inherent limitation can be avoided, however, by pairing a non-destructive whole-rock analytical method with acid maceration. X-ray tomographic microscopy (μCT) is well-suited to such an approach, allowing compositional and textural differences within a sample to be imaged as a grayscale representation of density variation in three dimensions, revealing fossil material within. By then subjecting the sample to acid maceration, individual fossils can be resolved and measured before and after chemical preparation, and any breakage or loss documented and quantified.

Herein, we employ a paired approach combining μCT-generated three-dimensional volume data of bulk rock samples with investigation of subsequently acid-extracted insoluble residues from the same scanned samples to test the hypothesis that smaller fossils would be disproportionately less likely to survive preparation and be recovered from acid-digested residues and to further assess and quantify introduced biases and their implications for paleoecological studies. For this study, we used prepared cores of carbonate bulk rock samples from the Edinburg Formation of Virginia, USA, which is well-known to host abundant and diverse silicified Ordovician marine fossils (Whittington & Evitt, 1953; Kraft, 1962). The pervasive silicification of the fossils within this unit makes it amenable to delivering sufficient contrast between the skeletal material and the host matrix using μCT.

Materials and Methods

Geological setting

The Edinburg Formation is an Upper Ordovician unit of massively bedded black limestone with occasional shale interbeds, representing deep ramp to basinal deposition at the northwest of the Taconic foreland basin system after drowning of shallower ramp carbonate facies including the underlying Lincolnshire Limestone (Holland & Patzkowsky, 1996). K-bentonites derived from Taconic volcanism are sporadically present in the sediments of the foreland basin and have been sampled for radiometric ages, most notably the Millbrig bed, which lies up section of the Edinburg in the Martinsburg Formation and has a U-Pb date of 452.86 Ma (Mitchell et al., 2004; Sell, Ainsaar & Leslie, 2013).

All samples in this study were collected from the Liberty Hall facies of the Edinburg Formation exposed at Strasburg Junction rail cut, a well-studied site in the Shenandoah Valley of Virginia (Fig. 1, Table 1; Cooper & Cooper, 1946; Whittington & Evitt, 1953; Read, 1980; Jacobs & Carlucci, 2019). Hand sample observations revealed two starkly different lithologies (Table 1). Rocks from Horizon 1, approximately 0.5 m above the contact between the Lincolnshire Limestone and Edinburg Formation, were coarse-grained, with a sparry or dismicritic texture likely indicative of recrystallization during early diagenesis. Sparse fossils and possible intraclasts visible on fresh surfaces were weakly aligned with the direction of original bedding, suggesting directional sorting during deposition. Horizons 3, 4, 5, and 6 cropped out over 30 m up section (Table 1) and were fine-grained with only occasional sparry or hematitic grains. Fossils, where visible on fresh or weathered surfaces, were oriented randomly to original bedding and comprised arthropods, brachiopods, and occasional crinoid ossicles. Rocks from these horizons also displayed heavy rinds on weathered surfaces, frequently stained rusty orange to pale yellow with iron oxides/oxyhydroxides.

Figure 1 Location map and generalized stratigraphy of samples analyzed in this study.

Adapted from Jacobs & Carlucci (2019).

Table 1 Summary of horizons and samples used in this study.

Horizon	Latitude	Longitude	Core samples from horizon	Notes	
EB18-01	38.9969°N	78.3748°W	C1, C2, C3, C4	~0.5 m above contact between Lincolnshire LS and Edinburg Fm	
EB18-03	38.9968°N	78.3744°W	C1, C3, C4	~20–30 m up section from EB18-01	
EB18-04	—	—	C1, C2, C3	0.5 m up section from EB18-03	
EB18-05	—	—	C1, C4	1.0 m up section from EB18-04	
EB18-06	—	—	C2, C3, C4	~5 m up section from EB18-05	

Fossil extraction

From five horizons with fossils visible on weathered surfaces, 15 samples were prepared as cylindrical cores (Table 1), approximately 2 cm in diameter and ranging from 1 to 4 cm in height, normal to bedding. Each core was imaged via X-ray tomographic microscopy (μCT) at the X-ray Microanalysis Laboratory at the University of Missouri using a Zeiss Xradia 510 Versa X-ray microscope with an isotropic voxel size of approximately 30 μm, producing three-dimensional renders of the core interiors. All μCT scans were processed using Dragonfly software Build 941–v.4.2.2 for Windows, Object Research Systems (ORS) Inc, Montreal, Canada, 2018 (http://www.theobjects.com/dragonfly). Post-image processing within the software was conducted to reduce imaging artifacts using the Ring Removal and Median filters. Regions of high brightness (corresponding to high-density ferrous material) and low brightness (corresponding to low-density siliceous material) relative to the local background were thresholded and manually segmented, excluding regions at the very top and bottom of cores due to persistent boundary artifacts. Due to radial and longitudinal variation in background brightness across the cores, multiple brightness thresholds were used to segment siliceous material in some cores; in such cases, the total siliceous volume is the union of the volumes from segmentation by the various thresholds. Segmented volumes for ferrous and siliceous material were refined by removing small (<100 voxel total volume) islands and were then partitioned into distinct regions-of-interest (ROIs), using six-connectivity (voxels considered to be connected if sharing faces rather than only edges or vertices) for both purposes. Individual fossils and other objects were identified among these ROIs by visual inspection in 3-D representations of voxels and against individual 2-D projections of μCT imagery. In cases where a fragmentary or incompletely resolved fossil fell into multiple non-connected ROIs, those ROIs were merged into one; in cases where a single ROI contained multiple contiguous objects, it was manually partitioned.

After each core had been imaged with μCT, they were macerated in 10% acetic acid to dissolve calcareous material until fully disaggregated and no longer visibly evolving gas bubbles. Acetic acid was chosen over alternatives such as hydrochloric acid or formic acid to allow for slow dissolution and avoidance of fossil damage. Insoluble residues were washed with water and sonicated in 30-s intervals, iterating until the supernatant ran clear. This washing process was then repeated using Calgon solution (0.052 M Na6(PO3)6, 0.286 M NaHCO3) as a deflocculant, sonicating as before, until all loose clay was removed. Cleaned residues were then washed over a 250-μm sieve, oven dried, and picked for identifiable fossils.

Statistical analyses

Following scanning and segmentation, each object resolved in the μCT data was measured for minimum, maximum, and mean three-dimensional Feret diameter. Fossils that could be confidently identified to at least the phylum level were scored for additional taxon-specific features and anatomical measurements (Table 2).

Table 2 Anatomical measurements and classifications by taxon and sclerite.

Measurements taken from μCT and residue imagery for each of several common taxa. The main sclerites of trilobites (excluding hypostomes and librigenae) were assigned different sets of measurements due to fundamental differences in structure. Lengths, widths, and heights are measured in mm; angles are measured in degrees; other category variables were recorded as Booleans.

Taxon	Measurements and classifications	
Trilobita	Cranidium	Maximum width (tr.) of occipital lobe*, maximum width (tr.) between eyes**, total length (sag.)	
Thoracic segment	Maximum width (tr.) of axial ring, width (tr.) between fulcra of left and right pleurae, length (sag.) of axial ring	
Pygidium	Maximum width (tr.) of first axial ring, total length (sag.)	
Bryozoa	Growth form: dendroid or fenestrate	
Gastropoda	Presence/absence of outer walls of whorls, total height of shell, height of last complete whorl, width of last complete whorl, half-angle of teleoconch***	
Ostracoda	Body length (a.-p.), body height (d.-v.), articulation/disarticulation, presence/absence of lateral spine	
Notes:

* Raphiophorids have an effaced prosopon with indistinct furrows, making identification of the occipital lobe difficult in μCT especially. For this family of trilobites, occipital width was instead measured as the full width (tr.) of the sclerite from gena to gena at the occiput.

** For eyeless trilobites of the families Metagnostidae and Raphiophoridae, interocular distance was excluded.

*** This is the angle formed at the apex between the axis of coiling and a line tangent to the outer walls of the body whorls.

Fossils recovered from residues were photographed using a GIGAmacro Magnify2 Robotic Imaging System with Canon EOS Rebel T8i DSLR and Nikon T1 1× and 3× objectives. For large specimens, additional photographs were taken under a reflected light microscope (Nikon SMZ1500 tethered to a Nikon D600 DSLR) to record features not visible in top-down view. Photographs were processed and analyzed with FIJI/ImageJ software, using the Trainable Weka Segmentation plugin to isolate fossils against the image background (Andreola et al., 2004; Schindelin et al., 2012). Each fossil was measured for minimum and maximum two-dimensional Feret diameter as well as taxon-specific measurements corresponding to those taken from μCT data (Table 2).

For each core, both the segmented μCT render and acid residues were counted for the total abundance of fossil specimens. Trilobites were identified to the family level based on their general geometry, furrow pattern, and, in residues, their prosopon. Trilobite material not reliably assignable to a single family was treated as a separate category. Prosopon and other textural features were not easily resolvable in the μCT dataset, and so were not generally considered for taxonomic assignment in counts based on μCT results. Bryozoans were morphologically classified by growth form either as thin-branching, thick-branching, or fenestrate. Ostracods, gastropods, and bivalves were not further classified due their generally small sample size and coarse silicification, precluding the reliable identification of taxonomically relevant features.

Using R statistical software, datasets were subjected to NMDS in three dimensions based on the Bray-Curtis dissimilarity index, using the metaMDS function provided in the vegan package (Dixon, 2003; R Core Team, 2021). Sites with no counted fossils were excluded from the analysis. Further paleoecological analysis of taxon abundances within and between samples was performed using PAST statistical software, calculating ecological dominance within samples (Simpson’s D) and assessing compositional similarity between samples using the Bray-Curtis index (Hammer & Harper, 2001, 2022).

Results

A total of 582 distinct objects were resolved via μCT, 460 as a low-opacity siliceous phase and 122 as a high-opacity ferrous phase. Of these, 241 siliceous objects and 14 ferrous objects were identifiable to at least a coarse taxonomic level, with the remaining 219 and 108 respectively left unidentified (Fig. 2). Siliceous fossils were dominated by trilobites (n = 225), of which slightly less than half (n = 99) could be confidently assigned to a family classification; including Asaphidae (c.f. Isotelus), Cheiruridae (Ceraurus), Metagnostidae (Trinodus), Pterygometopidae (Calyptaulax), Raphiophoridae (Ampyx, Lonchodomas), and Remopleurididae (Remopleurides), all of which are previously known from this site (Whittington & Evitt, 1953; Evitt, 1961). Non-trilobite siliceous material consisted of thin-branching and fenestrate bryozoans and a single valve from an ostracod. Ferrous fossils were mostly gastropods, with two bivalves, one possible thin-branching bryozoan, and one infilling of a raphiophorid cranidium (Table 3). This cranidium was counted towards siliceous and ferrous μCT object totals due to being preserved as silicified cuticle filled in by moldic pyrite but was counted as one individual in taxon totals (Fig. 2D). Resulting µCT TIFF stacks for each core are available at the MorphoSource link: https://www.morphosource.org/projects/000546507?locale=en. All DOIs are available just before the References section.

Figure 2 μCT renderings of fossils identified in cores of the Edinburg Formation.

Blue indicates silica and red and pink indicate Fe-rich mineralization. (A) Fenestrate bryozoan (01-C4-001). (B) Nuculanid bivalve (03-C3-064) with three small gastropods (03-C3-060, 03-C3-062, 03-C3-065) in pink. (C) Closer view of gastropod (03-C3-060). (D) Raphiophorid cranidium preserved in silica (05-C4-005) with pyrite infilling (05-C4-047). (E) Pterygometopid cephalon (04-C2-028). Note the compound eyes. (F) Agnostid fragment (06-C4-027). (G) Pterygometopid pygidium (05-C1-019). All scale bars approximate 1 mm.

Table 3 Fossils recovered by preparation method and taxon.

Total counts of all objects identified in μCT and residue imagery, grouped by taxon. Trilobites are further broken down to the family level. Trilobite material lacking sufficient anatomical features to be confidently assigned to a family was given its own category.

Taxon	µCT	Residues	
Total Trilobita	225	594	
Trilobita	Asaphidae	6	20	
Cheiruridae	2	3	
Metagnostidae	6	14	
Odontopleuridae	0	18	
Pterygometopidae	6	41	
Raphiophoridae	57	149	
Remopleurididae	22	59	
Unknown Trilobita	126	290	
Bivalvia	2	1	
Brachiopoda	0	5	
Bryozoa	Thin branching	9	304	
Thick branching	0	49	
Fenestrate	7	22	
Gastropoda	10	11	
Ostracoda	1	232	
Unknown	327	161	
Total	581	1,379	

Residues yielded 1,349 recognizable objects, 1,222 preserved as silica, 125 as ferrous minerals (pyrite, hematite, and/or limonite), and the remaining two as other indeterminate materials (neither were identifiable as a fossil) (Fig. 3). Nearly all siliceous fossils were attributed to at least a coarse taxonomic level, with only 48 unidentifiable. Conversely, 113 of the ferrous objects could not be identified. Siliceous objects in residue were made up largely of trilobites, fragments of thin-branching bryozoans, and ostracods. All trilobite genera represented in μCT were found also in residues, along with trilobites from the family Odontopleuridae (c.f. Ceratocephala). The remainder consisted of thick-branching and fenestrate bryozoans along with occasional brachiopod fragments. Ferrous fossils were exclusively represented by gastropods, except for a single bivalve (Table 3).

Figure 3 Photomicrographs and SEM images of fossils recovered in macerate residues.

(A) Silicified fenestrate bryozoan (01-C1). (B) Silicified erect branching bryozoan (03-C3). (C) Pyritized gastropod (03-C3). (D) Silicified raphiophorid cranidium (03-C1). (E) Silicified Calyptaulax cranidium (06-C4). (F) Silicified agnostid fragment (06-C4). (G) Silicified Calyptaulax pygidium (03-C1). (H) SEM-image of silicified ostracod valve (04-C1). (I) SEM-image of silicified trilobite pygidium (04-C1).

Tests of preparation bias

A contingency table was constructed containing μCT identified specimens counts, broken down by taxonomic grouping and by presence or absence in residues. The relationship between taxon and recovery was investigated using Pearson’s χ2 test (Table 4). The only taxonomic group which deviated significantly (p = 0.04) from overall likelihood of recovery was Unidentified Trilobita—material clearly from trilobites but lacking anatomical features in μCT sufficient to assign it to a family—which, despite being identified in μCT, are disproportionately unlikely to be recovered in residues.

Table 4 Taxonomic effects on recovery.

Contingency table containing counts of fossils with distinctive geometry identified in μCT imagery and either recovered or not recovered in residue. Residuals of recovered counts are standardized to the expected number of recovered specimens for a taxon; positive residuals indicate disproportionately high likelihood of recovery in residues, while negative values indicate lower-than-average likelihood of recovery. Statistically significant results (p < 0.05) are in bold.

Taxon	Not recovered	Recovered	Recovered residual	p-value	
Asaphidae	3	2	0.213	0.84	
Cheiruridae	1	1	0.285	0.67	
Metagnostidae	2	4	1.855	0.11	
Pterygometopidae	4	1	−0.787	0.46	
Raphiophoridae	37	20	−0.376	0.90	
Remopleurididae	13	8	0.493	0.81	
Unknown Trilobita	71	28	−7.389	0.04	
Bivalvia	1	1	0.285	0.67	
Bryozoa (thin)	4	4	1.140	0.39	
Bryozoa (fenestrate)	2	4	1.855	0.11	
Gastropoda	4	6	2.425	0.10	

Application of Pearson’s χ2 test to total counts of taxa identified in μCT and in residue found strong support (p = 1.89 × 10−35) for different proportional abundances in the two preparation types (Table 5). Raphiophorids (p = 1.99 × 10−5), remopleuridids (p = 0.02), unidentified trilobites (p = 9.92 × 10−17), bivalves (p = 0.02), and gastropods (p = 2.08 × 10−4) are significantly more abundant in μCT, while thin-branching bryozoans (p = 3.26 × 10−14), thick-branching bryozoans (p = 1.15 × 10−3), and ostracods (p = 1.27 × 10−13) are significantly more abundant in residues.

Table 5 Comparison of taxonomic abundances.

Contingency table containing total counts of taxa identified in μCT and in residue imagery. Residuals of recovered counts are standardized to the expected number of recovered specimens for the combination of taxon and preparation; positive μCT residuals indicate overrepresentation in μCT for that taxon, while positive residue residuals indicate overrepresentation in residues (relative to average proportional abundance for all fossils). Statistically significant results (p < 0.05) are in bold.

Taxon	μCT	Residue	μCT residual	Residue residual	p-value	
Asaphidae	6	20	0.715	−0.326	0.43	
Cheiruridae	2	3	1.224	−0.559	0.18	
Metagnostidae	6	14	1.372	−0.627	0.13	
Odontopleuridae	0	18	−1.762	0.805	0.05	
Pterygometopidae	6	41	−0.741	0.338	0.41	
Raphiophoridae	57	149	3.598	−1.643	2.0 E−05	
Remopleurididae	22	59	2.146	−0.980	0.02	
Unknown Trilobita	126	290	6.399	−2.922	9.9 E−17	
Bivalvia	2	1	2.060	−0.941	0.02	
Brachiopoda	0	5	−0.929	0.424	0.31	
Bryozoa (thin)	9	304	−6.125	2.797	3.3 E−14	
Bryozoa (thick)	0	49	−2.908	1.328	1.2 E−3	
Bryozoa (fenestrate)	7	22	0.892	−0.407	0.32	
Gastropoda	10	11	3.350	−1.530	2.1 E−4	
Ostracoda	1	232	−6.183	2.824	1.3 E−13	

The presence of size bias in the likelihood of recovery was evaluated using the Mann-Whitney U-test with a null hypothesis of no difference in medians and a standard threshold of significance of α = 0.05 (Mann & Whitney, 1947). In comparisons between fossils identified in both μCT and residue and those identified in μCT but not in residue, there was no statistically significant difference of maximum (p = 0.54), minimum (p = 0.09), or mean (p = 0.24) Feret diameters, nor of elongation factor (p = 0.16) defined as the ratio of maximum to minimum Feret diameters. However, when testing overall distributions of size between all fossils identified in μCT and all those found in residue, the U-test found a dramatic difference in maximum Feret diameter (p = 1.75 × 10−79) between the medians of the two groups (3.98 mm for μCT, 1.41 mm for residue).

Exploratory paleoecological ordinations

NMDS performed separately on the μCT (Table 6) and residue (Table 7) datasets resulted in ordinations with stress scores of 0.06 and 0.08, respectively (Fig. 4). Taxon loadings in both ordinations consistently grouped bivalves and gastropods nearby each other. Ordinations of both μCT and residues tended to group medium-sized benthic trilobites (raphiophorids, pterygometopids, and cheirurids) together along with unidentified trilobite material (most of which likely derived from one of those families), with pelagic trilobites (remopleuridids) plotting closer to gastropods and bivalves. Asaphids, likely represented here by the extremely large benthic trilobite Isotelus, consistently fall near the latter cluster but are represented only by fragments of cuticle. The ordination position of agnostids, whose life habit remains controversial (Fortey & Owens, 1999), is inconsistent between μCT and residues, falling near the benthic and pelagic trilobite clusters in those analyses, respectively. In both cases, they are closely accompanied by ostracods (which can occupy pelagic or benthic niches and whose life habit was not interpreted in this study). Due to the extreme disparity between identification in μCT and recovery in residues of ostracods, their location within the ordination for μCT data is likely not informative.

Table 6 Abundance of taxonomic groups identified in μCT data used in NMDS.

	01-C1	01-C2	01-C4	03-C1	03-C3	03-C4	04-C1	04-C2	04-C3	05-C1	05-C4	06-C1	06-C3	06-C4	Sum	
Agnostida	0	0	0	1	0	0	1	2	0	1	0	0	0	1	6	
Asaphidae	0	0	0	0	0	1	2	0	1	1	1	0	0	0	6	
Cheiruridae	0	0	0	0	0	0	0	0	0	1	1	0	0	0	2	
Odontopleuridae	0	0	0	0	0	0	0	0	0	0	0	0	0	0	0	
Pterygometopidae	0	0	0	0	0	0	1	1	0	3	0	0	0	1	6	
Raphiophoridae	0	0	0	3	1	0	5	4	5	23	14	0	0	2	57	
Remopleuridae	0	0	1	0	5	1	2	1	5	4	1	0	2	0	22	
Trilo indet	2	0	1	8	12	0	19	15	16	31	5	2	6	9	126	
Bivalvia	0	0	0	0	1	0	0	0	0	0	0	0	1	0	2	
Brachiopoda	0	0	0	0	0	0	0	0	0	0	0	0	0	0	0	
Bryo thin	3	1	2	1	0	0	0	0	0	0	1	0	0	1	9	
Bryo thick	0	0	0	0	0	0	0	0	0	0	0	0	0	0	0	
Bryo fenestrate	1	0	6	0	0	0	0	0	0	0	0	0	0	0	7	
Gastropoda	0	0	0	0	3	0	2	1	0	0	0	0	4	0	10	
Ostracoda	0	0	0	0	0	0	0	1	0	0	0	0	0	0	1	
Sum	6	1	10	13	22	2	32	25	27	64	23	2	13	14		

Table 7 Abundance of taxonomic groups recovered in acid maceration residues used in NMDS.

	01-C1	01-C2	01-C3	01-C4	03-C1	03-C3	03-C4	04-C1	04-C2	04-C3	05-C1	05-C4	06-C1	06-C3	06-C4	Sum	
Agnostida	0	0	0	0	6	2	0	2	1	0	0	0	0	2	1	14	
Asaphidae	0	0	0	0	0	2	1	4	1	2	0	0	0	4	6	20	
Cheiruridae	0	0	0	0	1	0	0	0	0	1	0	1	0	0	0	3	
Odontopleuridae	0	0	0	0	9	0	0	4	1	0	3	0	0	0	1	18	
Pterygometopidae	0	0	0	0	12	4	0	5	2	0	1	5	1	4	7	41	
Raphiophoridae	0	0	0	0	59	8	1	21	7	2	19	12	2	10	8	149	
Remopleuridae	0	1	0	0	8	30	0	6	2	2	2	0	0	4	4	59	
Trilo indet	2	8	0	2	72	51	6	22	17	8	37	12	8	29	16	290	
Bivalvia	0	0	0	0	0	1	0	0	0	0	0	0	0	0	0	1	
Brachiopoda	0	0	0	1	2	0	0	0	1	0	1	0	0	0	0	5	
Bryo thin	42	81	0	43	17	24	0	0	6	3	11	13	0	36	28	304	
Bryo thick	8	8	4	15	4	0	1	0	0	0	0	0	0	7	2	49	
Bryo fenestrate	10	0	0	9	2	0	0	0	0	0	0	0	0	0	1	22	
Gastropoda	0	0	0	0	0	6	0	0	0	0	0	0	0	5	0	11	
Ostracoda	0	3	1	0	57	15	6	110	13	10	9	4	0	2	2	232	
Sum	62	101	5	70	249	143	15	174	51	28	83	47	11	103	76		

Figure 4 Results of NMDS of taxon abundance data for (A) μCT data (stress = 0.06) and (B) acid maceration residue data (stress = 0.08).

Samples are plotted as grey squares. Taxon scores are indicated by blue circles. Convex hulls contain samples from each sampling horizon (Table 1). Red: EB18-01. Orange: EB18-03. Yellow: EB18-04. Green: EB18-05. Blue: EB18-06.

Thin-branching and fenestrate bryozoans plotted relatively close together, forming an isolated cluster. Thick-branching bryozoans were not closely associated with the other two morphotypes in the residue ordination and were not detected in μCT.

Samples from individual horizons tended to fall near each other in loose association. Two main clusters are apparent in both residue and μCT ordinations: one characterized by low NMDS1 scores and dominated by bryozoans, and one with higher NMDS1 scores and dominated by arthropods (trilobites and ostracods).

For each sample, dominance was computed for both μCT (Table 6) and residue (Table 7) taxon totals, with 95% confidence intervals based on 9999 bootstrap replicates (Fig. 5). The results of this analysis differed notably between the two; μCT dominance values fell in a tight band between 0.25 and 0.40 (excepting two outliers with n < 3), while residue dominance values had a bimodal distribution with one group of values between 0.40 and 0.65 and another falling between 0.15 and 0.30 (Fig. 5). Rarefaction curves were calculated for both μCT (Table 6) and residue (Table 7) counts of each sample using PAST (Hammer & Harper, 2022; Fig. 6). Curves were visually inspected for the presence of an inflection point as a rough qualitative assessment of sampling completeness; a sample’s curve “leveling off” (sudden decrease in slope) is considered informal evidence that taxa in the true population are well-represented in the sample (Sanders, 1968; Raup, 1975). This inflection point was observed in residue counts from most samples (prominently in 03-C1, 03-C3, 04-C1, and 06-C3), but not in μCT counts. This suggests that the smaller total counts of individuals in μCT data are leading to the non-recovery of rarer taxa. Such under-sampling is a liability of ecological analyses based on those samples.

Figure 5 Simpson’s dominance (D) and bootstrapped 95% confidence intervals calculated for each sample by data type.

µCT data are in blue and acid maceration residue data are in orange.

Figure 6 Rarefaction analysis of taxonomic richness for (A) μCT samples (Table 6) and (B) acid maceration residue samples (Table 7).

Cluster analysis

Multivariate cluster analyses were performed in PAST to provide alternate metrics of similarity in the paleoecological composition of the samples, using the Bray-Curtis index as before (Bray & Curtis, 1957; Hammer & Harper, 2022). Dendrograms with branch length scaled to similarity were constructed for μCT counts, residue counts, and a combined dataset treating μCT and residue counts for each sample as two separate sites (Figs. 7 and 8). Cluster analysis of μCT counts grouped samples from Horizon 04, and treated Horizon 05 similarly, but scattered samples from Horizons 03 and 06 across the tree. Using residue data, Horizon 05 formed a cluster as before but Horizon 04 split, with cores 04-C2 and 04-C3 remaining close but 04-C1 further removed; Horizons 03 and 06 were dispersed as in the μCT tree. In both trees, samples from Horizon 01 were far removed from other samples, tending to form a cluster basal to the rest of the samples.

Figure 7 Bray-Curtis similarity dendrograms of taxon abundance values in (A) μCT samples (Table 6) and (B) acid maceration residue samples (Table 7) calculated separately.

Figure 8 Bray-Curtis similarity dendrograms of pooled taxon abundance values in μCT samples (Table 6) and acid maceration residue samples (Table 7).

The combined tree retains the general close groupings of the two individual trees due to sharing the same dissimilarity metric (Fig. 8). Notably, even for Horizons 01, 04, and 05 whose samples tended to cluster in both the μCT and residue trees, those clusters are not closely related in the overall analysis; there are few clear patterns in this tree, but μCT samples tend to cluster more closely with other μCT samples, and residue samples with residue samples, than do μCT and residue counts of the same sample or even of samples from the same horizon.

Discussion

Size and taxon effects on recovery

We hypothesized at the outset of this study that body size would be negatively associated with likelihood of recovery in residues; larger fossils may be more susceptible to breakage, and larger organisms with thicker skeletal elements may not fully silicify, producing a brittle outer husk. However, the results of our analysis do not support this possible effect. While fossils recovered in residue have a smaller median body size than those resolved in μCT, this appears to be due not to preferential breakage of larger fossils during washing but rather disproportionate non-resolution of smaller fossils in μCT datasets. Rather than μCT acting as a baseline to test the biases of acid maceration based on body size, the results of this study suggest the opposite.

Overrepresentation with respect to the abundance of pyritized taxa (gastropods and bivalves) in µCT may be the result of the high contrast between iron-bearing phases and the matrix, making such fossils easier to resolve in μCT than otherwise-comparable siliceous fossils. Alternatively, because most of the pyritized fossil materials represent internal molds, they may be harder to identify outside of the 3D context provided in μCT, where they remain within their shell. If the latter is the case, this likely reflects lower fidelity of moldic pyrite preservation than that of the replacive silicification. Results of the NMDS ordinations consistently plotted molluscan taxa nearby one another, which may reflect either a shared infaunal environment or instead early burial conditions conducive to pyritization in those deposits. However, the low abundance of bivalves (2 in μCT, 1 recovered from residues, always alongside multiple gastropods) makes this apparent association tenuous. It is noteworthy that disseminated pyritization in the form of non-fossil granules was present in several samples (e.g., 05-C4) that contained no mollusks, suggesting that the presence or absence of mollusks in a sample may reflect their abundance in the paleoenvironment and is not solely controlled by whether early burial chemistry allowed pyritization to proceed. Ostracods, meanwhile, are disproportionately overrepresented in residue due to their small size making them difficult to resolve in μCT at the resolution used in this study. Branching bryozoans also suffer from this, but their numbers in residue are likely inflated by fragmentation of large individuals into many smaller ones.

Nonrecovery bias and potential causes

Revealing that the number of unidentified trilobite fossils recognized in μCT is significantly less likely to be recovered in residue is to be expected. Fragmentation and incomplete preservation are common causes for trilobite material to lack identifying characteristics, and both are likely to promote further breakage and degradation during washing by compromising the structural integrity of the sclerite. This may also influence likelihood of recovery by impeding identification of the fossil in residue; a fragment of cuticle lacking identifying characteristics is less likely to be recognized as corresponding to an object observed in μCT, and minor breakage is likely to disrupt recognizable aspects of its outline and other key features.

Diagenetic biases arise from a variety of sources and are heavily dependent on shell microstructure, organic matter, and availability of reactive and replacive ions to the shell. It is well-documented that certain taxa are more susceptible to silicification than others, and, even within taxa, textural differences can make certain skeletal elements more likely to be preserved than others (Daley & Boyd, 1996; Cherns et al., 2011; Butts, 2014). Organic matter content within a shell also influences the likelihood of silicification by providing nucleation sites for the deposition of silica, though the chemistry of this is complex (Wallace, DeYoreo & Dove, 2009; Butts, 2014). Similarly, replacement via pyrite tends to initiate at sites of organic matrix in the shell but is also dependent upon reducing microenvironments and/or microbial zonation within the sediment (Fisher, 1986; Canfield & Raiswell, 1991; Schiffbauer et al., 2014).

Further biases can arise from the sample extraction and preparation process itself. During washing, larger fossils may be tumbled against other grains or container walls, leading to abrasion or breakage, while smaller fossils may be crushed beneath larger objects as grains settle. Elongated grains may be more prone to breakage than spheroidal grains of the same volume due to a lower minimum cross-sectional area, which can lead to taxon or sclerite bias against fossils with rod-like geometry, such as branching bryozoans and the spines of some trilobites. Breakage during preparation can cause taxa to be undercounted due to indiscriminate destruction of individuals or damage sufficient to remove recognizable features. Paradoxically, it can also inflate taxon counts by turning one fossil into many still-recognizable fragments. These overlapping effects can produce a wide variety of biases in size and shape, and the cumulative effect on measurements of abundance cannot reasonably be predicted a priori.

The composition of the host rock can further impose preparation-related biases. For instance, well-cemented rocks frequently require longer maceration periods than poorly consolidated ones, while argillaceous rocks may need more thorough washing to drive off insoluble clays, often including sonication. The longer sediment is washed and manipulated, the more breakage tends to occur, making matrix texture an important factor of preparation bias. While the acid maceration processes can be accelerated by using alternative acids, such as hydrochloric, the violent effervescence caused by the intensity of the reaction can also have a deleterious effect on the extraction of delicate forms.

Artifacts and limitations of µCT

While a powerful tool for visualization, μCT does present some unavoidable sources of potential error: artifacts arising from specimen capture and processing, particularly the “hardening” of the X-ray beam through preferential absorption of lower-energy photons by the sample. Beam hardening is a ubiquitous issue in μCT, but not all hardening artifacts are of equal impact. Since the beam is slightly hardened by passing through the outer surface of the sample, the interior is always slightly darkened relative to the outermost layer; these “cupping artifacts” are produced predictably based on sample geometry and can be corrected relatively easily (Schladitz, 2011; Jung et al., 2011; Abel, Laurini & Richter, 2012).

Heterogeneous samples may experience additional artifacts. Regions of higher-density material (e.g., iron minerals in sediments, bones in biological samples) can drastically harden beams passing through them, causing dark blotches to appear around bright features within a sample and especially in the spaces between multiple bright features. These artifacts are far more difficult to correct due to their irregular shape and remain the subject of ongoing research in economic geology and other materials science fields (Remeysen & Swennen, 2006; Park, Chung & Seo, 2015; Bam et al., 2019). Given the abundant pyrite in the material examined in this study, this stands as a caveat to the results reported here; similar methods applied to rocks with lower density variation may yield clearer μCT data and therefore more complete identification of fossils within the samples. Filtering during pre-processing, or correction applications in post-processing software packages, can reliably improve these artifacts, but often at the cost of introducing noise along the axis of sample rotation (Kyriakou, Prell & Kalender, 2009; Yousuf & Asaduzzaman, 2009).

Relevance for paleoecological interpretation

Broad categories of biofacies are generally consistent between residue- and μCT-derived taxon abundances as interpreted using NMDS. While total counts for μCT are much lower than those of residues, relative abundances in the former largely recapitulate the latter with some notable exceptions; ostracods are almost absent from μCT data, due to resolution constraints imposed from the diameter of the cores, even when highly abundant in residues, and bryozoan counts tend to be much higher in residues than in μCT due to breakage transforming one large fragment into many small fragments. For this reason, the application of μCT data alone to describe paleoecological structure should be undertaken with caution. Ordinations used in paleoecology are sufficiently complex that such biases in recovery can, by introducing error and uncertainty upstream, irreparably taint conclusions drawn downstream.

Overall, the use of μCT counts on their own to characterize silicified remains in these limestone biofacies is of uncertain value. Sample volume constraints make bulk samples unfeasible, and the cost of instrument time can be a roadblock to replication and large-scale sampling. However, μCT may be best suited when chemical preparation is dangerous or impractical due to matrix or fossil mineralogy, or when studying sponges, bryozoans, corals, or other modular organisms for which fragmentation during washing can obscure or inflate the number of individuals present. Samples best suited to the approach outlined in this study will be densely fossiliferous (mitigating the limitation of sample volume), bear taxa with morphological features conducive to identification in μCT (related to shape rather than texture) and contain grains more susceptible to dissolution than the matrix or cement joining them together. Such rocks should also have a contrast in density, whereby the material of interest has a higher X-ray attenuation, for better visualization, whilst keeping in mind the potential effects of beam hardening artifacts. Conodonts and other phosphatic fossils, for example, are occasionally known from silicified sediments, and characterizing taxonomic assemblages using μCT techniques may constitute an alternative to digestion from the toxic and hazardous hydrofluoric acid normally used to extract fossils from siliceous cements (Green, 2001).

Conclusions

The main goal of this study was to quantify the effects of taxonomic affinity and body size on the likelihood of fossil recovery as revealed by different extraction methods. Breakage remains an unavoidable concern with acid maceration, one which μCT can provide insight to, though not without its own caveats. While μCT imaging of silicified fossils in limestone matrix can resolve morphological features of interest, the issues faced in this specific study may limit this technique’s ability to answer broader paleoecological questions. Analyzing original rock contents as a means of detecting breakage during preparation may be valuable as a control on fragmentary abundance counts, but limitations of sample size and the potential for taxonomic bias to affect ordinations present serious pitfalls to analysis or characterization of biofacies. It can, however, be useful in establishing broad categories if not finer gradations between related assemblages. Since the drawbacks resulting from beam hardening and other μCT artifacts are heavily dependent on the properties of the sample, these methods may prove more effective when applied to rocks containing different fossil taxa or with different lithologic compositions from the materials studied here.

Variation in fossil recovery can stem not only from easily observable sedimentological features such as grain size and composition but also from redox chemistry and solute profiles of pore fluids during burial and early diagenesis, which can be challenging to infer from samples without more in-depth geochemical analyses. Original shell composition, both in terms of organic content and aragonitic vs calcitic (high- or low-magnesium) mineralogy is likely relevant due to its influence on the spontaneity and kinetics of silicification chemistry. Information on the breadth of variation is limited but suggests a wide range of possible taxonomic outcomes based on sample lithology.

We gratefully acknowledge L.G. Huntley for assistance with fieldwork and William Foster, Agata Jurkowska, and an anonymous reviewer for their helpful commentary and suggested improvements.

Additional Information and Declarations

Competing Interests

Author Contributions

Field Study Permissions

Data Availability

The authors declare that they have no competing interests.

Gabriel S. Jacobs conceived and designed the experiments, performed the experiments, analyzed the data, prepared figures and/or tables, authored or reviewed drafts of the article, and approved the final draft.

Sarah Monique Jacquet conceived and designed the experiments, analyzed the data, authored or reviewed drafts of the article, and approved the final draft.

Tara Selly conceived and designed the experiments, performed the experiments, analyzed the data, authored or reviewed drafts of the article, and approved the final draft.

James D. Schiffbauer conceived and designed the experiments, analyzed the data, authored or reviewed drafts of the article, and approved the final draft.

John Warren Huntley conceived and designed the experiments, analyzed the data, prepared figures and/or tables, authored or reviewed drafts of the article, and approved the final draft.

The following information was supplied relating to field study approvals (i.e., approving body and any reference numbers):

The samples were collected on public land and no permits were required.

The following information was supplied regarding data availability:

The CT Images are available at MorphoSource:

https://doi.org/10.17602/M2/M546524 (Core Eb18 01 C4);

https://doi.org/10.17602/M2/M546518 (Core Eb18 01 C2);

https://doi.org/10.17602/M2/M546513 (Core Eb18 01 C1);

https://doi.org/10.17602/M2/M546540 (Core Eb18 03 C4);

https://doi.org/10.17602/M2/M546543 (Core Eb18 03 C3);

https://doi.org/10.17602/M2/M546542 (Core Eb18 03 C1);

https://doi.org/10.17602/M2/M546581 (Core Eb18 04 C1);

https://doi.org/10.17602/M2/M546571 (Core Eb18 04 C2);

https://doi.org/10.17602/M2/M546576 (Core Eb18 04 C3);

https://doi.org/10.17602/M2/M546603 (Core Eb18 06 C4);

https://doi.org/10.17602/M2/M546601 (Core Eb18 06 C3);

https://doi.org/10.17602/M2/M546600 (Core Eb18 06 C1);

https://doi.org/10.17602/M2/M546602 (Core Eb18 05 C4);

https://doi.org/10.17602/M2/M546599 (Core Eb18 05 C1).

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
