# Peer review of "Resolving taphonomic and preparation biases in silicified faunas through paired acid residues and X-ray microscopy"

_PeerJ, doi:10.7717/peerj.16767_

## Round 0.1 · original submission · Major Revisions

Dear Gabriel,

I have now received the comments from the reviewers on your paper. The comments are thorough, and I believe it will greatly improve your paper. The main issues raised by the reviewers are the statistics and the geological and microscopic background. I do not believe these issues will be difficult to deal with; I am looking forward to the revised paper.

·

Basic reporting

no comment

Experimental design

no comment

Validity of the findings

no comment

Additional comments

This manuscript aims to investigate how studying silicified fossil assemblages might be subject to certain biases that overall affect our interpretations of past community structures. It is a nice and tidy manuscript and as scientists become more concerned about taphonomic effects on our interpretations, articles like this one will serve as important contributions to the literature. Obviously, this manuscript doesn't study all points of bias, but it is also not expected to.

I do not really have any criticisms or worthwhile suggestions that would improve the manuscript. There are lots of tiny quirky things, but it is not all that important. E.g., referencing software is all over the place, different fonts, citation style; personally less than 1 mm is not very small in my view; I don't understand why you would use R and Past software, if you can program in R do the whole analysis in R and make that code publicly available for transparency; p values should be reported consistently not p <0.000001 vs. p <0.05; I found placed holder sentences like acknowledging reviewers AB&C disingenuous when the article is finally published. But none of these issues stop me from recommending this for publication.

·

Basic reporting

The article fulfill sth eimportant field of sample preparation as well as its taphonomy. The literature need carreful checking. The professional language especially in terms of geology (lithilogy) needs to be checked.

Experimental design

The artcile, altough presents very well the labolatory and analytical studies, filed in geological description of the study area. The lithological column with precise location of a samples should be added. Moreover the process of a silification should be discussed more precisely as a general. Also the section with macro and microskopic (thisn section analysis, SEM-EDS analysis) descripition of a samples should be added.

Validity of the findings

The atricle add valuse as a good recipy for sample preparation.

Additional comments

The article should be improved by adding an good geological background of a sampling, fieldworks and microscopic descriptions.

Reviewer 3 ·

Basic reporting

.

Experimental design

.

Validity of the findings

.

Additional comments

This manuscript compares the acid macerated silicified fossils with micro-CT scans of the same fossils, considering a number of potential causes for variation between the datasets such as taxon, size, and fragmentation due to processing.

It is an elegant manuscript and a valuable contribution to the literature, and I recommend publication. Some general comments and minor suggestions follow.


General comments:

It is cleanly and clearly written, and it does an excellent job of walking the reader through the logic of its experimental design.

The stats are thorough and well explained. I particularly appreciate the level of background given in their reporting – while perhaps a bit out of the realm of ‘results’, this gives the reader the ability to understand the authors’ rationality for making the comparisons they’re making, and makes it much more readable/interpretable.

Multiple biases are considered – e.g. processing, visual interpretation by observer of images, mineral type, host rock, identifiability, fragmentation, initial and final size, and likely habitat.

It’s particularly interesting that acid maceration is recommended, based on their results, as a baseline to microCT, since microCT is often held up as the gold standard whose use is only precluded by time and money.



Suggestions:

Although mentioned obliquely in another part of the text, it might be good to state in the Methods section why acetic acid was used rather than HCl.

Similarly, it’s specified in the Discussion, and sort of in the Methods, but since it’s so vital to the paper maybe say exactly how fossils were numerated in the methods; e.g. they were included if could be identified to Order level, not just as fossil; pieces of same animal were identified as individual fossils, etc. The use of ‘object’ and ‘fossil’ is a bit unclear in first two paragraphs of the results sections.

In the graphical figures the yellow color is a bit hard to see – could it be darkened or something?

Typos:

- line 142 ‘… to test’

-line 351 parentheses are outside of sentence

- line 393 ‘similarly also’ seems a bit redundant?

---

## Round 0.2 · Minor Revisions

Thank you for addressing the comments of the reviewers.

The only outstanding matter is access to the data. Please mention in the text that the raw data will be available through an online repository in the manuscript (including permanent link: doi or arc link in case of Morphosource). You have providing links during the review but it is currently not mentioned in the manuscript.

all best,
Shaw Badenhorst

·

Basic reporting

The artcile is clear, very detailed and properly focused on the arguments.

Experimental design

Origuinal reserach, very helpful in paleontological studies.

Validity of the findings

The studies has an impact in terms of sample preparation adn paleontological studies.

Additional comments

The Authors adressed all my comments and suggestions. The article has been improved.

---

## Round 0.3 · accepted · Accept

Thank you for adding the link to the data in the manuscript. I am pleased to inform you that the paper has now been accepted for publication.